# Quantification of Gleason Pattern 4 at MRI-Guided Biopsy to Predict Adverse Pathology at Radical Prostatectomy in Intermediate-Risk Prostate Cancer Patients

**DOI:** 10.3390/cancers15225462

**Published:** 2023-11-17

**Authors:** Hubert Kamecki, Łukasz Mielczarek, Stanisław Szempliński, Małgorzata Dębowska, Paweł Rajwa, Michael Baboudjian, Jakob Klemm, Juan Gómez Rivas, Elza Modzelewska, Omar Tayara, Wojciech Malewski, Przemysław Szostek, Sławomir Poletajew, Piotr Kryst, Roman Sosnowski, Łukasz Nyk

**Affiliations:** 1Second Department of Urology, Centre of Postgraduate Medical Education, 01-809 Warsaw, Poland; 2Nałęcz Institute of Biocybernetics and Biomedical Engineering, Polish Academy of Sciences, 02-109 Warsaw, Poland; 3Department of Urology, Comprehensive Cancer Center, Medical University of Vienna, 1090 Vienna, Austria; 4Department of Urology, Medical University of Silesia, 41-800 Zabrze, Poland; 5Department of Urology, North Hospital, AP-HM, 13007 Marseilles, France; 6Department of Urology, University Medical Center Hamburg-Eppendorf, 20246 Hamburg, Germany; 7Department of Urology, Hospital Clinico San Carlos, 28040 Madrid, Spain; 8Department of Urology and Oncological Urology, Warmian-Masurian Cancer Center, 10-228 Olsztyn, Poland

**Keywords:** prostate cancer, prostate biopsy, Gleason pattern 4, risk stratification

## Abstract

**Simple Summary:**

The amount of Gleason pattern 4 (GP4) in biopsy material may be used for individual risk stratification in prostate cancer patients. The utility of this parameter is potentially most significant in patients that fall into the intermediate-risk category, in whom various alternative strategies are under consideration. We aim to assess the performance of multiple methods of GP4 quantification in predicting the features of advanced disease at final surgery. Our retrospective analysis of data from 123 patients who underwent magnetic-resonance imaging-guided biopsy and radical prostatectomy revealed that the commonly used method of risk assessment with GP4 amount relative to cancer length may be a poor predictor of high-risk disease, as compared to other quantification methods, including our newly developed concept of GP4 volume. The results of this study may serve as the basis for further research aimed at refining the risk-assessment strategies in prostate cancer.

**Abstract:**

Background: Data on Gleason pattern 4 (GP4) amount in biopsy tissue is important for prostate cancer (PC) risk assessment. We aim to investigate which GP4 quantification method predicts adverse pathology (AP) at radical prostatectomy (RP) the best in men diagnosed with intermediate-risk (IR) PC at magnetic resonance imaging (MRI)-guided biopsy. Methods: We retrospectively included 123 patients diagnosed with IR PC (prostate-specific antigen <20 ng/mL, grade group (GG) 2 or 3, no iT3 on MRI) at MRI-guided biopsy, who underwent RP. Twelve GP4 amount-related parameters were developed, based on GP4 quantification method (absolute, relative to core, or cancer length) and site (overall, targeted, systematic biopsy, or worst specimen). Additionally, we calculated PV×GP4 (prostate volume × GP4 relative to core length in overall biopsy), aiming to represent the total GP4 volume in the prostate. The associations of GP4 with AP (GG ≥ 4, ≥pT3a, or pN1) were investigated. Results: AP was reported in 39 (31.7%) of patients. GP4 relative to cancer length was not associated with AP. Of the 12 parameters, the highest ROC AUC value was seen for GP4 relative to core length in overall biopsy (0.65). an even higher AUC value was noted for PV × GP4 (0.67), with a negative predictive value of 82.8% at the optimal threshold. Conclusions: The lack of an association of GP4 relative to cancer length with AP, contrasted with the better performance of other parameters, indicates directions for future research on PC risk stratification to accurately identify patients who may not require immediate treatment. Incorporating formulas aimed at GP4 volume assessment may lead to obtaining models with the best discrimination ability.

## 1. Introduction

Since the development of the original Gleason scale, our understanding of the role of particular prostate cancer (PC) morphology patterns has been changing in line with contemporary evidence. Cancer consisting purely of Gleason pattern 3 (GP3) is known to be of non-significant malignant potential [1,2] and subject to active surveillance (AS) [3]. On the other hand, tissues harboring features of Gleason pattern 4 (GP4) are associated with an elevated risk of progression [4], and it has been demonstrated that greater amounts of GP4 correspond to a more aggressive disease [5]. Thus, PC composed of both GP3 and GP4 is classified into two distinct grade groups (grade group (GG) 2 and GG3, according to International Society for Urologic Pathology (ISUP)), based on the relative predominance of one of the patterns [6].

It has been demonstrated that the absolute quantity of GP4 in biopsy specimens may better indicate the risk of adverse pathology (AP) at radical prostatectomy (RP) than the relative amount [7]. The GP4 length at biopsy has been also shown to be strongly associated with the risk of biochemical recurrence after RP [8]. The clinical importance of these considerations is high, given the potential role of AS in selected patients with low amounts of GP4 [3]. However, the interpretation of biopsy results may be highly dependent on sampling variation, as well as heterogeneous approaches to GP4 amount reporting [9]. As most of the available evidence on the relationship between GP4 amount and risk of AP is based mainly on data from patients who underwent template ultrasound-guided prostate sampling, updating this evidence in the era of magnetic resonance imaging (MRI)-guided biopsy is necessary.

The goal of this study is to assess patients diagnosed with intermediate risk (IR) PC at MRI-guided biopsy and to provide comprehensive data on the role of different GP4 quantification methods in predicting AP at RP.

## 2. Materials and Methods

We retrospectively analyzed consecutive patients who underwent RP in our institution (Second Department of Urology, Centre of Postgraduate Medical Education, Warsaw, Poland) between May 2018 and January 2023, and included those who: (i) underwent MRI-guided biopsy of the prostate at our institution and (ii) were diagnosed with GG 2–3 cancer at biopsy. Exclusion criteria were: (i) extraprostatic extension (EPE) or seminal vesicle invasion (SVI) at pre-biopsy MRI, (ii) pre-biopsy serum PSA ≥ 20 ng/mL, or (iii) missing or incomplete data in regard to the analyzed variables. Data were collected from medical patient records and included: age, pre-biopsy PSA, pre-biopsy prostate MRI characteristics, biopsy pathology report, and radical prostatectomy pathology report. All data were anonymized for the purpose of this study.

### 2.1. MRI-Guided Biopsy

All biopsies were performed at our institution and were either cognitive MRI-guided or MRI-ultrasound software fusion biopsies. The cognitive biopsies were performed with a transrectal approach, using an “end-fire” probe. The fusion biopsies were grid-based and performed via either a transperineal or transrectal approach. All biopsies were performed or supervised by a highly experienced physician. In all biopsy-naïve patients, systematic cores were included. In patients with a previous negative biopsy, a systematic biopsy might have been omitted. The number and distribution of cores were at the discretion of the performing physician. Systematic cores typically do not cover the regions subject to targeted biopsy.

### 2.2. Biopsy Pathology Reporting

All the specimens were assessed by an experienced urogenital pathologist. Reports were provided in accordance with the ISUP/World Health Organization (WHO) guidelines. At biopsy, multiple cores were gathered into single specimens if collected from the same site: separate specimens for each targeted lesion and up to two specimens for systematic biopsy (a single specimen for each lobe). Therefore, pathology reports did not include data regarding individual cores and each specimen was assessed as a whole. Every report included: total length of cores (mm), relative total cancer length and Gleason pattern 4 length (% of total core length), data on cribriform architecture, among other specimen features.

### 2.3. Gleason Pattern 4 Quantification Methods

Firstly, we calculated the amount of GP4 as either a percentage of the total core length (GP4%cores), percentage of the total cancer length (GP4%cancer), or absolute length in mm (GP4mm). Secondly, we calculated the amounts of GP4 for overall biopsy (OB, which included all cores), targeted biopsy (TB, all specimens with targeted cores), systematic biopsy (SB, all specimens with systematic cores), or the worst specimen (WS, the specimen either from a targeted lesion or a systematically biopsied lobe that contained the highest amount of GP4). This led to the development of 3 × 4 = 12 GP4 quantification methods.

As the pathology reports provided Gleason pattern 4 length as a percentage (rounded to 1%) of the total length of cores in a specimen (rounded to 0.1 mm), we had to recalculate the Gleason pattern 4 length in millimeters.

Also, we developed a novel parameter aimed to represent the total volume of GP4 in the prostate, i.e., PV × GP4, calculated as prostate volume (PV) × GP4%cores/OB.

### 2.4. Radical Prostatectomy and Adverse Pathology

All patients underwent either traditional laparoscopic or robot-assisted radical prostatectomy. The decision to perform extended lymph node dissection (eLND) was primarily based on the calculated preoperative risk of lymph node involvement (LNI). Adverse pathology (AP) in the final specimen was defined as either: (i) EPE or SVI (T3a or higher), (ii) GG 4 or 5, and (iii) LNI.

### 2.5. Outcome Measurements and Statistical Analysis

Categorical and quantitative variables were calculated as numbers with percentages and medians with interquartile ranges, respectively. Patients were divided into AP and no-AP groups and compared. Associations between the analyzed variables and a dependent variable were investigated using logistic regression models and the outcomes were expressed as odds ratios (ORs) with 95% confidence intervals (95% CIs). For the selected variables, receiver-operator characteristic (ROC) curves were developed to demonstrate the ability to discriminate between AP or no AP. The areas under the curves (AUCs) were calculated and expressed together with asymptotic 95% CIs. For diagnostic performance analysis, the optimal threshold was defined according to the Youden index.

The results were considered statistically significant at a *p*-value < 0.05. Statistical analyses were performed using MATLAB version R2023a (MathWorks, Natick, MA, USA).

## 3. Results

We identified 178 patients with GG2 or GG3 PC at biopsy who underwent both MRI-guided biopsy and RP at our institution in the analyzed period. We excluded 35 patients due to EPE/SVI at pre-biopsy MRI, 12 patients due to PSA ≥ 20 ng/mL, and 8 patients due to an incomplete biopsy pathology report. Eventually, 123 patients were included in the analyses (see Appendix A). The characteristics of the included patients are presented in Table 1.

AP was found in 39 (31.7%) patients at RP. The comparison between AP and non-AP patients is presented in Table 2.

In univariable logistic regression models (Appendix A), the variables with significant differences at AP vs. no-AP comparison (Table 2) maintained statistically significant associations with AP, except for PSA (*p* = 0.060), and no previously non-significant variable became significant. Despite using forward and backward stepwise selection, we failed to develop a multivariable logistic regression model with more than one statistically significant variable, probably due to multiple, highly significant interdependencies between the analyzed factors, as demonstrated in the Spearman’s rank correlation analysis (Appendix A).

Variables with statistically significant differences at AP vs. no-AP comparison were further selected for ROC/AUC analysis. The AUC measures, together with the optimal threshold values and negative predictive values (NPVs) for AP, are presented in Table 3.

## 4. Discussion

To our knowledge, this is the first comprehensive investigation of the associations between GP4 quantification methods at MRI-guided biopsy and AP at RP in IR PC patients. We introduced the concept of GP4 volume, proposing PV × GP4 as a novel parameter to assess this measurement. In our patients, PV × GP4 demonstrated the highest ability to discriminate between AP and no AP, with values <2.55 mL being highly negatively predictive (82.8%) of AP (Table 3). Of the traditional GP4 quantification methods, the highest ROC AUC value was seen for GP4%cores/OB (Table 3). Interestingly, the performance of GP4%cancer, the method established as the basis for the contemporary grade group classification system, was poor.

Over 20 years ago, Chan et al. were the first to document that differentiation between Gleason score (GS) 3 + 4, and GS 4 + 3 was independently associated with progression risk after RP [10]. In 2012, Reese et al. demonstrated that the further subclassification of GS 3+4/4+3 patients into six groups, based on the GP4 amount relative to cancer tissue, led to a better prediction of the final pathology results and the risk of biochemical relapse (BCR) [11]. Multiple further studies proved that the detailed reporting of GP4 percentage at biopsy provides more diagnostic or prognostic information than the simple differentiation between GG2 and GG3 [12,13,14,15,16,17,18,19,20,21]. While most of the researchers investigated the significance of GP4 percentage relative to cancer tissue, the promising performance of other quantification methods, including GP4 percentage relative to overall biopsied tissue [22,23] or absolute GP4 length [7,8], has been demonstrated.

Our results contribute to the contemporary discussion on expanding AS inclusion criteria into IR PC. At present, offering AS to highly selected IR PC patients is allowed by the guidelines [3], with one of the criteria being a GP4 amount of <10% (relative to cancer length), a threshold first established as a recommendation in the 2015 Canadian consensus [24]. Our study provides new insight into this topic. Firstly, we demonstrated that, in contemporarily diagnosed IR patients, a risk assessment based on GP4%cancer (and GG2 vs. 3 differentiation) may not represent the true aggressiveness of the disease the best, and other GP4 quantification methods should be considered. Secondly, as most of the previous evidence was based on data from patients who underwent traditional template biopsy, our results may better address the evidence gap in the era of MRI-guided biopsy.

The lack of significant difference in GP4%cancer between AP and no-AP patients is not surprising, as GP4%cancer is dependent on the amount of GP3. While GP3 is indolent [1,2], its quantification for the purpose of PC risk stratification may be misleading. Furthermore, GP4%cancer provides no data representing the absolute size of clinically significant PC (csPC) burden. The better performance of other GP4 quantification methods in our patients is in line with other emerging evidence [7].

Of the GP4-non-related parameters, only PSA demonstrated an association with AP in our patients. Prostate Imaging Reporting and Data System (PIRADS) category 5 rates were not different between AP and no-AP; however, it should be noted that EPE/SVI cases were excluded from the analyses, and thus PIRADS 5 vs. 3–4 differentiation was only lesion size-dependent. The most intriguing finding was the lack of association between the presence of cribriform architecture at biopsy and AP at RP, which is contrary to contemporary evidence [25]. One explanation is that our analysis was performed among the selected population of IR patients only. Also, the quantification of cribriform tissue, instead of present or absent dichotomization, might have better represented the risk profile of a PC patient [19]. Lastly, this might have been caused by a small sample size.

While our assessment of GP4 volume is a novel approach, tumor volume (TV) has already been shown to predict AP at RP in GG2 and GG3 PC patients [18]. A parameter obtained by multiplying TV and GP4%cores/TB could have represented an even more refined method to assess the GP4 volume than PV × GP4. However, we were limited by the lack of data regarding TV.

The presented associations between PV × GP4 with AP must be interpreted with caution, and the use of PV × GP4 in clinical practice should be avoided until higher-quality evidence corroborates our findings. Nevertheless, we believe that the development and refinement of parameters representing GP4 volume could be a promising direction for future research on risk assessment in IR PC.

All patients underwent MRI-guided prostate sampling, which we consider a strength of the study. On the other hand, it may be difficult to extrapolate our results into patients undergoing template-only biopsy, especially those with a negative MRI, as csPC still may be diagnosed in such a setting [26].

In our cohort, the biopsies did not follow a standardized institutional protocol. As a result, the findings should be interpreted with the awareness of the potential risk of inter-procedure sampling variability. While taking an unusually low or high number of targeted or systematic cores could influence the analyzed parameters, we believe that this had a minor effect in our population, as all the procedures were performed or supervised by two highly experienced physicians. Nevertheless, given that MRI-guided biopsy is inherently a personalized procedure, a nonuniform sampling might more accurately represent real-world scenarios.

Most of the limitations to our study result from its retrospective character. Our sample was relatively small, which might have caused the lack of differences between AP and no-AP groups for several parameters, as well as the inability to develop a multivariable model. We lacked survival data and, while AP at RP may be considered predictive of worse prognosis, no direct insight into the relationship between GP4 amounts and survival outcomes makes the interpretation of our results limited. Specimens were not reviewed; however, the original exams were performed by experienced urogenital pathologists and reported using a uniform institutional protocol. Both cognitive and fusion biopsy patients were included, which might have led to selection bias, as reports of cognitive biopsy being associated with an inferior tumor sampling have been published in the literature [27]. A total of 3 out of 123 patients did not undergo systematic biopsy, which might have slightly influenced some of the reported parameters. As eLND was performed in selected cases only, LNI and, thus, AP rates might have been underestimated; however, as pN1 was reported only in four (3.1%) cases, all of them already having been associated with pT3 disease, the role of this bias was most probably minor, if any. Other biopsy features, known for their association with worse outcomes, e.g., lymphovascular invasion [28] or perineural invasion [29], were not analyzed, which may be considered another limitation. As GP4mm was not directly provided in pathology reports, we recalculated this parameter, which might have caused a minor risk of bias due to the rounding of the values. Also, the biopsy reports lacked pathology data for single cores, which, if available, might have allowed for including additional GP4 quantification methods in the analyses. Lastly, no exclusion criteria in regard to biopsy quality were implemented; however, given the high median total length of cores in our patients, we do not consider this a significant issue.

## 5. Conclusions

We provided comprehensive data on the role of multiple GP4 quantification methods in predicting AP in patients diagnosed with IR PC at MRI-guided biopsy. We demonstrated the lack of a significant difference in GP4%cancer values between AP and no-AP patients. This result, when contrasted with the better performance of the other parameters, suggests a need for further research to develop superior risk stratification methods based on GP4 quantification, with the ultimate goal of accurately identifying IR PC patients who may not require immediate treatment. Our findings suggest that incorporating formulas aimed at GP4 volume assessment, e.g., PV × GP4, may lead to obtaining models with the best discrimination ability. Given the several limitations, the results of this study should be interpreted with caution.

## Figures and Tables

**Table 1 cancers-15-05462-t001:** Characteristics of the included patients.

Characteristic			All Patients (*n* = 123)
Median age, years (IQR)			66 (62–71)
Median PSA, ng/mL (IQR)			7.4 (5.7–9.4)
Maximum PIRADS category			
	2		4 (3.3%)
	3		11 (8.9%)
	4		71 (57.7%)
	5		37 (30.1%)
Median PV, mL (IQR)			34 (29–47)
Median PSAD, ng/mL^2^ (IQR)			0.21 (0.13–0.28)
Type of MRI-guided biopsy			
	cognitive		61 (49.6%)
	software fusion		62 (50.4%)
Number of targeted lesions			
	1		73 (59.3%)
	2		38 (30.9%)
	≥3		12 (9.8%)
Median length of cores, mm (IQR)			
	overall		132 (99–172)
	targeted		64 (44–98)
	systematic ^a^		64 (39–82)
Grade group at biopsy ^b^			
	2		97 (78.9%)
	3		26 (21.1%)
GP4 amount, median (IQR)			
	GP4%cores ^c^	OB	5.8% (3.3–10.6%)
		TB	10.0% (5.0–20.0%)
		SB	0.0% (0.0–1.2%)
		WS	15.0% (8.0–25.0%)
	GP4%cancer ^c^	OB	30.9% (20.1–43.7%)
		TB	33.3% (22.2–50.2%)
		SB	7.1% (0.0–32.1%)
		WS	36.4% (26.5–54.4%)
	GP4mm ^c^	OB	7.0 (3.9–14.5)
		TB	6.9 (4.0–12.2)
		SB	2.2 (1.5–3.3)
		WS	6.5 (3.6–12.0)
	PV × GP4, mL ^c^		2.51 (1.11–3.61)
Cribriform pattern present			59 (48.0)
Grade group at RP			
	1		5 (4.1%)
	2		59 (48.0%)
	3		46 (37.4%)
	4		13 (10.6%)
	5		0 (0.0%)
Tumor stage at RP			
	T2		95 (77.2%)
	T3a		22 (17.9%)
	T3b		6 (4.9%)
pN1			4 (3.3%)
Adverse pathology			39 (31.7%)

IQR, interquartile range; PSA, prostate-specific antigen; PIRADS, Prostate Imaging Reporting and Data System; PV, prostate volume; PSAD, PSA density; MRI, magnetic resonance imaging; GP4, Gleason pattern 4; OB, overall biopsy; TB, targeted biopsy; SB, systematic biopsy; WS, worst specimen; RP, radical prostatectomy. ^a^ Systematic biopsy was omitted in 3 patients. ^b^ Based on the overall biopsy Gleason pattern 4 and 3 prevalence. ^c^ See text for explanation.

**Table 2 cancers-15-05462-t002:** Comparison between patients with and without adverse pathology features at radical prostatectomy.

Characteristic			AP (*n* = 39)	No–AP (*n* = 84)	*p*-Value
Median age, years (IQR)			67 (64–72)	65 (60–71)	0.158
Median PSA, ng/mL (IQR)			8.9 (6.2–10.4)	7.1 (5.1–8.9)	0.029
PIRADS category 5			13 (33.3%)	24 (28.6%)	0.745
Median PV, mL (IQR)			38 (30–52)	34 (28–42)	0.078
Median PSAD, ng/mL^2^ (IQR)			0.21 (0.15–0.28)	0.21 (0.13–0.28)	0.587
Grade group 3 (vs 2) ^a^			11 (28.2%)	15 (17.9%)	0.191
GP4 amount, median (IQR)					
	GP4%cores ^b^	OB	8.7% (5.1–12.1%)	5.2% (3.0–9.3%)	0.009
		TB	13.2% (6.4–22.4%)	10.0% (5.0–20.0%)	0.248
		SB	0.0% (0.0–2.4%)	0.0% (0.0–0.0%)	0.031
		WS	20.0% (10.0–30.0%)	11.5% (6.0–20.0%)	0.027
	GP4%cancer ^b^	OB	33.3% (22.1–52.8%)	30.3% (20.0–40.0%)	0.129
		TB	40.0% (25.8–58.8%)	33.3% (20.4–42.9%)	0.062
		SB	22.5% (0.0–34.8%)	0.0% (0.0–26.1%)	0.089
		WS	40.0% (33.3–61.9%)	34.3% (22.2–44.6%)	0.054
	GP4mm ^b^	OB	9.9 (5.1–19.5)	6.5 (3.9–10.9)	0.033
		TB	9.0 (4.9–17.9)	6.4 (4.0–10.0)	0.047
		SB	2.3 (1.2–3.0)	1.8 (1.6–3.7)	0.799
		WS	8.7 (4.8–14.7)	6.0 (3.3–9.7)	0.044
PV × GP4, mL			2.95 (1.82–4.14)	1.85 (0.94–3.27)	0.003
Cribriform pattern present			19 (50.0%)	40 (47.6%)	0.962

AP, adverse pathology (see text for explanation); IQR, interquartile range; PSA, prostate-specific antigen; PSAD, PSA density; PIRADS, Prostate Imaging Reporting and Data System; PV, prostate volume; GP4, Gleason pattern 4; OB, overall biopsy; TB, targeted biopsy; SB, systematic biopsy; WS, worst specimen. ^a^ Based on the overall biopsy Gleason pattern 4 and 3 prevalence. ^b^ See text for explanation.

**Table 3 cancers-15-05462-t003:** Receiver-operator characteristic areas under the curves for the selected variables, discriminating between adverse pathology and no adverse pathology at radical prostatectomy.

Variable	ROC AUC (95% CI)	Optimal Threshold	NPV, % (95% CI)
GP4%cores, OB	0.65 (0.56–0.74)	6.4%	80.3 (70.7–89.9)
GP4%cores, SB	0.60 (0.50–0.69)	– ^a^	– ^a^
GP4%cores, WS	0.62 (0.53–0.71)	12.0%	79.2 (68.3–90.2)
GP4 mm, OB	0.62 (0.53–0.71)	11.3 mm	76.2 (67.1–85.3)
GP4 mm, TB	0.62 (0.52–0.71)	9.9 mm	76.9 (67.6–86.3)
GP4 mm, WS	0.61 (0.52–0.70)	8.1 mm	75.7 (65.9–85.5)
PV × GP4	0.67 (0.58–0.75)	2.55 mL	82.8 (73.6–92.1)

ROC AUC, receiver-operator characteristic area under the curve; CI, confidence interval; NPV, negative predictive value; GP4, Gleason pattern 4; OB, overall biopsy; TB, targeted biopsy; SB, systematic biopsy; WS, worst specimen; PV, prostate volume. ^a^ Not calculated due to non-significant ROC AUC.

## Data Availability

The data analyzed in this study are available upon request from the corresponding author.

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
