# Peer review of "Quantification of Gleason Pattern 4 at MRI-Guided Biopsy to Predict Adverse Pathology at Radical Prostatectomy in Intermediate-Risk Prostate Cancer Patients"

_cancers, 2023, doi:10.3390/cancers15225462_

Round 1
Reviewer 1 Report
Comments and Suggestions for Authors
Congratulations on the present study! It is well-designed and structured and debates an interesting subject.
There are a few aspects that I suggest being improved and I will state this point by point:
1. Please extend a bit the introduction section.
2. Please explain if all the patients received informed consent. Also, it would be useful to state if the Declaration of Helsinki protocol was respected.
3. Do the authors think including only fusion biopsies would be more appropriate? Do you think the results would be the same after excluding cognitive biopsies?
4. You can include a table for inclusion and exclusion criteria?
5. For LND did you include the Briganti nomogram?
6. Please include an abbreviation list for this study.
Comments on the Quality of English Language
Minor English spelling issues
Author Response
Dear Reviewer,
We would like to express our appreciation for your effort to review the manuscript and for the valuable comments you provide. Below we respond to your specific queries.
- Please extend a bit the introduction section.
Response: Thank you for raising our attention at this issue. In order to better introduce the reader into the topic we expanded the second paragraph of this section, so that most important findings of the cited studies are provided (lines 66-69).
- Please explain if all the patients received informed consent. Also, it would be useful to state if the Declaration of Helsinki protocol was respected.
Response: As our study was retrospective and non-interventional, and data was anonymized, the institutional review board statement and patient informed consent were not mandatory, which was in accordance with national law regulations. For the same reasons, the Declaration of Helsinki does not apply to our study. In the Materials and Methods, we now clarified that all data was anonymized. The ethical statements are provided at the footnotes section of the manuscript.
- Do the authors think including only fusion biopsies would be more appropriate? Do you think the results would be the same after excluding cognitive biopsies?
Thank you very much for this interesting point. In the literature, there are reports of cognitive biopsies being associated with inferior tumor sampling, as compared to software-fusion biopsies. Indeed, this might have led to selection bias in our study, and we mentioned this possible limitation in the last paragraph of our Discussion. However, it should be noted that still there is no consensus regarding the superiority of fusion biopsy and it may be the surgeon’s experience where much of the quality of the biopsy lies in. All biopsies in our patients were performed by highly experienced physicians (one physician for cognitive, and one for fusion biopsies) and there is no good reason to assume that the quality of the procedures was different in the cognitive vs fusion biopsy groups. However, the two physicians might have adhered to different biopsy protocols. While this potential bias was unavoidable, we believe that it better represents the real-world scenario. We mentioned this limitation it in the revised version of the manuscript (lines 241-245). As the sample size would be too small if we split it into two halves (cognitive or fusion biopsy patients), any differences most likely would be non-significant.
- You can include a table for inclusion and exclusion criteria?
Thank you for this advice. In order to present this information in an even more attractive way, we drew a flowchart and attached it to the manuscript (Supplementary Figure 1).
- For LND did you include the Briganti nomogram?
Thank you for this important question. As mentioned in section 2.4. (Radical prostatectomy and adverse pathology), eLND was performed based on preoperative risk of lymph node involvement. In our institution, the Briganti nomogram is typically used for this purpose, however, we cannot exclude that the MSKCC calculator was used in some cases. We typically perform eLND if calculated risk of LNI is 7% or higher, however the final decision is also based on general clinical picture, age of the patient and comorbidities. We slightly clarified this in the revised version of the manuscript (lines 126-127). However, we believe that focusing on details of how decisions were made falls beyond the scope of this paper.
- Please include an abbreviation list for this study.
Thank you for this note. We agree that the abbreviation list would increase the readability of the text, however, this would be against the editorial policy of the journal.
Once again, please accept our gratitude. With the help of your suggestions, we have made changes to the manuscript, and we believe that its quality is now improved.
Reviewer 2 Report
Comments and Suggestions for Authors
The authors reported that in the era of MRI-guided biopsy, the percentage of GP4 in the prostate gland is a better predictor of adverse pathology than that of GP4 in the cancer. They also reported that a novel parameter, PVxGP4, showed an even better AUC.
The results of this study are interesting, but the sample size and methodology are inadequate for introducing a new approach to GP4 volumetry and do not appear to have the statistical power to influence the existing ISUP Gleason grade (GP4%cancer) evidence.
1. The detailed methods of biopsy (e.g., the number of biopsy cores, etc.) are not described. Due to the small sample size and the variability of the biopsy methods in the cohort, one might question the reliability of the respective parameters. Patients who did not undergo a systematic biopsy are also included, and in such cases, the GP4%core may be higher than the actual value. At the very least, these patients need to be excluded from the analysis. Additionally, patients with higher number of targeted biopsies also tend to have a higher GP4%core.
2. The AUC is reported to be less than 0.7. If tumor volume is considered important, MRI volumetry would likely be more useful since MRI is routinely performed. The superiority and clinical significance of PVxGP4 should be fully discussed.
Comments on the Quality of English LanguageWhat is meant by "internal MRI-guided biopsy (Line 82)"?
Author Response
Dear Reviewer,
We are sincerely grateful for the valuable time spent on reviewing our manuscript and for the important comments you provided.
You mention that our analysis does not carry enough statistical power to influence the existing grade group evidence. Thank you for this comment. We fully agree with this message. Please note, that in our manuscript we avoid judgmental wording and what we conclude is that our study suggests a need for further studies in this area. However, as our findings are in line with other, previously published results, this need seems to be well supported by the data. The concept of PVxGP4 is novel and we never considered it a core parameter analyzed in our study, our goal was rather to seek for new paths and directions for future research. Despite many advances in diagnostics, contemporary risk assessment in intermediate-risk prostate cancer is still far from being sufficient.
Below we provide our responses to your other comments.
- The detailed methods of biopsy (e.g., the number of biopsy cores, etc.) are not described. Due to the small sample size and the variability of the biopsy methods in the cohort, one might question the reliability of the respective parameters. Patients who did not undergo a systematic biopsy are also included, and in such cases, the GP4%core may be higher than the actual value. At the very least, these patients need to be excluded from the analysis. Additionally, patients with higher number of targeted biopsies also tend to have a higher GP4%core.
Thank you very much for highlighting the possible causes bias in our study. Those are very important comments, and we believe that they concisely describe the core limitations.
We tried to provide all details on how the biopsies were performed in the section 2.1. (MRI-guided biopsy). As we stated, the number and distribution of cores were at the discretion of the performing physician. Typically, more systematic cores were taken in cases of a larger prostate volume. Regarding targeted cores, the decision to take more cores might have been made during the biopsy, if the risk of missing is anticipated (e.g., the patient moved, or the lesion shape was challenging). Sadly, we do not have a uniform institutional biopsy protocol. We discuss it as a limitation in the revised version of the manuscript (lines 250-257).
Our relatively small sample size may be considered a limitation, and we mention it in the Discussion. However, as all the procedures were either performed (vast majority) or supervised (occasionally) by two highly experienced physicians (one for cognitive and one for fusion biopsies), we believe that the variability of the biopsy methods in the cohort was minor. Anyway, we mention this risk of inter-procedure sampling variability as possible cause of bias in the revised version of the paper (lines 250-257).
The inter-procedure sampling variability can inevitably lead to potential over- or underrepresentation of GP4 in the specimens. While adhering to a uniform biopsy protocol might have revealed more pronounced associations, it is worth noting that MRI-guided prostate biopsy is inherently a personalized procedure. Consequently, such uniform sampling might be infrequent in real-world scenarios.
Only three patients (out of the 123) did not undergo systematic biopsy (this data is provided in the footnotes to Table 1). While, indeed, GP4%cores/OB (as well as other OB-related parameters) might have been falsely increased in those cases, we believe that including them did not influence the overall results. Please note, that despite this limitation, the association between GP4%cores and AP demonstrated the best statistical significance of all twelve parameters. And removing those patients would further decrease the sample size, as well as lead to loss of valid data regarding other (TB-, SB- and WS-related) parameters. Moreover, such diversity, even if small, better represents the real-world scenario. Thank you very much for raising our attention at this issue. We are mentioning this limitation in the revised paper (lines 267-269).
- The AUC is reported to be less than 0.7. If tumor volume is considered important, MRI volumetry would likely be more useful since MRI is routinely performed. The superiority and clinical significance of PVxGP4 should be fully discussed.
No AUC in our study reached 0.7. Thank you very much for raising this issue. As AUC below 0.7 is generally considered indicative of inadequate discrimination ability, our results show that the proposed parameters are not yet suitable for predicting AP at RP. The diagnostic performance might have been better, e.g., with multivariable models involving factors that we lacked. We are not suggesting that we have provided the clinicians with a superior tool for risk assessment. Instead, we are emphasizing that the contemporary concept of assigning grade groups 2 and 3 may need re-evaluation. Our study presents insights into the potential roles of other parameters. Further prospective research is essential, and our findings offer evidence-based guidance for planning such studies.
Thank you for drawing our attention to the unclear role of PVxGP4. We do not suggest superiority of this parameter. While we cannot predict what the association would be, in the Discussion we have mentioned the limitation of not including tumor volumetry into the present study (lines 236-240). Nonetheless, we developed PVxGP4 to seek for novel methods to approach the concept of GP4 amount, and we demonstrate that the association of PVxGP4 with AP was even higher than in case of the twelve “traditional” parameters. There is no clinical significance of PVxGP4 at this point, as higher-quality evidence is required to allow for implementing this method into clinical practice. Nevertheless, our main message, supported by the example of PVxGP4, is that the development and refinement of parameters representing GP4 volume could be a promising direction for future research on risk assessment. We further discussed it in the revised version of the manuscript (lines 241-245).
Thank you very much for pointing out that the phrase “internal MRI-guided biopsy” sounds unclear. We revised it as: “MRI-guided biopsy of the prostate in our institution” (line 84).
Once again, thank you for your thorough review and invaluable feedback on our manuscript. Your insights have greatly enriched our work, and we deeply appreciate the time and expertise you have dedicated to this process.
Reviewer 3 Report
Comments and Suggestions for Authors
The manuscript discussed the utility of using Gleason Pattern 4 volume from MRI targeted biopsy as a way to predict the presence of adverse pathology at the time of RP as a way to risk stratifying patient considering active surveillance for intermediate risk prostate cancer. While I applauded the authors for the efforts to investigate a clinically significant dilemma, there are a few caveats of the current manuscript.
The authors developed multiple metrics involving GS4 absolute and relative length (regarding total cores). One important thing to consider is that absolute length of GS4 tumors (to a lesser extent the relative length) depends on how many cores were taken. From line 95 "the number ... of cores were at the discretion of the performing physician". The variation of number of cores obtained will cause significant variation of length of GS4 tumor detected unless it is normalized.
Additionally, the parameter PVxGP4 really did not make any biological sense. This is also affected by the total length of biopsy cores (i.e. how many cores were taken) and it does not represent the total volume of disease.
These flaws from the study design unfortunately render conclusions from the study irrelevant.
Author Response
Dear Reviewer,
We would like to express our sincere gratitude for taking the time to review our manuscript and provide your valuable insights. Your expertise is essential in ensuring the quality and rigor of our work, and we genuinely appreciate your contribution to this process.
Thank you for bringing to our attention the potential influence of variability in core sampling on our results. In our institution, we do not have a uniform biopsy protocol, i.e., both the number of targeted and systematic cores may differ between patients. For example, in case of larger prostates more systematic cores may be taken. In cases of lesions difficult to access or increased risk of missing the target due to patient movements, targeted biopsy may include more cores, as well. While taking an unusually low or high number of cores in many patients might have significantly influenced the overall results in our patient group, we doubt that such variable sampling took place. Please refer to Table 1, which shows relatively narrow IQRs for overall, targeted, and systematic cores length (also, this data on core length may help in determining which patients our findings can be extrapolated to). All our biopsies were either performed (vast majority) or supervised (occasionally) by two highly experienced physicians (one for cognitive and one for fusion biopsies), and we believe that the variability of sampling in the cohort was minor.
The inter-procedure sampling variability can inevitably lead to potential over- or underrepresentation of GP4 in the specimens. While adhering to a uniform biopsy protocol might have revealed more pronounced associations, it is worth noting that MRI-guided prostate biopsy is inherently a personalized procedure. Consequently, such uniform sampling might be infrequent in real-world scenarios.
We further discussed the above limitations in the revised version of the manuscript (lines 250-257).
Thank you for highlighting your concerns about the role and clarity of PVxGP4. We are aware that the significance of this parameter may be uncertain. While the biological sense of PVxGP4 may appear to be minor, we developed this parameter to explore novel methods to approach the concept of GP4 amount and PVxGP4 appears to provide some insight into possible volume of GP4 in the prostate. It is evident that variable proportions of targeted vs systematic cores length, as well as variable amounts of GP4 in targeted vs systematic cores, cause significant bias to this approach to GP4 volume. However, please note that the observed association between PVxGP4 and AP showed the strongest statistical significance of all other methods (furthermore, GP4%cores in overall biopsy, susceptible to the aforementioned bias due to variable proportions of targeted vs systematic cores, demonstrated the most robust association among the twelve “traditional” parameters). Currently, PVxGP4 has no established clinical significance, as higher-quality evidence is required to allow for implementing this method into clinical practice. Nevertheless, our main message, supported by the example of PVxGP4, is that the development and refinement of parameters representing GP4 volume could be a promising direction for future research on risk assessment. We further discussed it in the revised version of the manuscript (lines 241-245).
We sincerely appreciate your thorough review and thoughtful feedback, which undoubtedly enhance the rigor of our work. Your expertise has been instrumental in refining our manuscript, and we are truly grateful for your time and contributions. We hope our revisions meet your expectations and look forward to any further recommendations or considerations.
Reviewer 4 Report
Comments and Suggestions for Authors
This is a retrospective analysis of the importance of quantification method of GP4 on biopsy MRI-guided PCa patients who underwent prostatectomy.
It is a well written paper that describe very clearly the methods and results, and most important the limitations of this work (discussion). It makes the reader to understand the (might) not significant relation founded. It is a good hypothesis that must continuous be explored in the future
Author Response
Dear Reviewer,
Thank you for dedicating your time and expertise to review our manuscript. Your thorough evaluation plays an indispensable role in ensuring the credibility and quality of our work. Even in the absence of specific comments, your confirmation of the manuscript's readiness is invaluable to us. We genuinely appreciate your commitment to upholding the standards of scholarly research.
Round 2
Reviewer 2 Report
Comments and Suggestions for Authors
The authors have responded sincerely to the reviewers' comments and revised the manuscript. Despite the limitations of the study design and its clinical significance, which the authors discuss in the Discussion, I believe that the paper is significant as it provides a direction for future research about Gleason grade group.
Reviewer 3 Report
Comments and Suggestions for Authors
Thank you for addressing the concerns I raised in my previous comments. No additional comments.